# Novel community health worker strategy for HIV service engagement in a hyperendemic community in Rakai, Uganda: A pragmatic, cluster-randomized trial

Larry W. Chang[1,2,3,4]*, Ismail Mbabali[4], Heidi Hutton[5], K. Rivet Amico[6], Xiangrong Kong[2,7], Jeremiah Mulamba[4], Aggrey Anok[4], Joseph Ssekasanvu[4], Amanda Long[3], Alvin G. Thomas[3], Kristin Thomas[3], Eva Bugos[3], Rose Pollard[3], Kimiko van Wickle[3], Caitlin E. Kennedy[3,4], Fred Nalugoda[4], David Serwadda[4], Robert C. Bollinger[1], Thomas C. Quinn[1,8], Steven J. Reynolds[1,4,8], Ronald H. Gray[2,4], Maria J. Wawer[1,2,4], Gertrude Nakigozi[4]

1 Division of Infectious Diseases, Department of Medicine, Johns Hopkins School of Medicine, Baltimore, Maryland, United States of America, 2 Department of Epidemiology, Johns Hopkins Bloomberg School of Public Health, Baltimore, Maryland, United States of America, 3 Department of International Health, Johns Hopkins Bloomberg School of Public Health, Baltimore, Maryland, United States of America, 4 Rakai Health Sciences Program, Rakai, Uganda, 5 Department of Psychiatry and Behavioral Sciences, Johns Hopkins School of Medicine, Baltimore, Maryland, United States of America, 6 Department of Health Behavior Health Education, University of Michigan, Ann Arbor, Michigan, United States of America, 7 Wilmer Eye Institute, Johns Hopkins School of Medicine, Baltimore, Maryland, United States of America, 8 Laboratory of Immunoregulation, Division of Intramural Research, National Institute for Allergy and Infectious Diseases, National Institutes of Health, Bethesda, Maryland, United States of America

* lchang8@jhmi.edu

**Data Availability Statement:** Data cannot be shared publicly due to the presence of participant identifiable data and because of the data sharing

## Abstract

### Background

Effective implementation strategies are needed to increase engagement in HIV services in hyperendemic settings. We conducted a pragmatic cluster-randomized trial in a high-risk, highly mobile fishing community (HIV prevalence: approximately 38%) in Rakai, Uganda, to assess the impact of a community health worker-delivered, theory-based (situated Information, Motivation, and Behavior Skills), motivational interviewing-informed, and mobile phone application-supported counseling strategy called "Health Scouts" to promote engagement in HIV treatment and prevention services.

### Methods and findings

The study community was divided into 40 contiguous, randomly allocated clusters (20 intervention clusters, *n* = 1,054 participants at baseline; 20 control clusters, *n* = 1,094 participants at baseline). From September 2015 to December 2018, the Health Scouts were deployed in intervention clusters. Community-wide, cross-sectional surveys of consenting 15 to 49-year-old residents were conducted at approximately 15 months (mid-study) and at approximately 39 months (end-study) assessing the primary programmatic outcomes of self-reported linkage to HIV care, antiretroviral therapy (ART) use, and male circumcision,

policies of the Rakai Health Sciences Program (RHSP, www.rhsp.org). However, a deidentified version will be provided to interested parties subject to completion of the RHSP data request form and signing of a Data Transfer Agreement. Interested parties may contact Dorean Nabukalu at datarequests@rhsp.org.

**Funding:** This work was supported by the National Institute of Mental Health (PI: LWC, R01MH107275), the Division of Intramural Research, the National Institute for Allergy and Infectious Diseases (TCQ, SJR), and the National Heart, Lung, and Blood Institute (AGT, T32HL007055), National Institutes of Health (www. nih.gov), and the Johns Hopkins University Center for AIDS Research (P30AI094189, hopkinscfar. org). The funders had no role in study design, data collection and analysis, decision to publish, or preparation of the manuscript.

**Competing interests:** I have read the journal's policy and the authors of this manuscript have the following competing interests: emocha Mobile Health Inc. developed and supported the smartphone application used in this study. LC and RB are entitled to royalties on certain non-research revenue generated by this company and own company equity. Specific to this study, LC and RB have and will receive no royalties or compensation from emocha Mobile Health Inc.. This arrangement has been reviewed and approved by the Johns Hopkins University in accordance with its conflict of interest policies. MW receive a consulting fee from the Rakai Health Sciences Program, which has been reported to the Johns Hopkins Conflict of Interest office. This consultancy did not influence the manuscript under review. RG reports personal fees from Rakai Health Sciences Program, outside the submitted work; and RG's contribution to this publication/presentation was as a member of the Board of Directors of the Rakai Health Sciences Program. This arrangement has been reviewed and approved by the Johns Hopkins University in accordance with its conflict of interest policies. All other study team members have declared that no competing interests exist.

**Abbreviations:** ART, antiretroviral therapy; CHI, combination HIV interventions; CHWs, community health workers; GPS, Global Positioning System; HTS, HIV Testing Services; mLAKE, mHealth Lakefolk Actively Keeping Engaged; PrEP, pre-exposure prophylaxis; PRR, prevalence risk ratios; RCCS, Rakai Community Cohort Study; RHSP, Rakai Health Sciences Program; sIMB, situated Information, Motivation, and Behavioral Skills.

and the primary biologic outcome of HIV viral suppression (<400 copies/mL). Secondary outcomes included HIV testing coverage, HIV incidence, and consistent condom use. The primary intent-to-treat analysis used log-linear binomial regression with generalized estimating equation to estimate prevalence risk ratios (PRR) in the intervention versus control arm. A total of 2,533 (45% female, mean age: 31 years) and 1,903 (46% female; mean age 32 years) residents completed the mid-study and end-study surveys, respectively. At mid-study, there were no differences in outcomes between arms. At end-study, self-reported receipt of the Health Scouts intervention was 38% in the intervention arm and 23% in the control arm, suggesting moderate intervention uptake in the intervention arm and substantial contamination in the control arm. At end-study, intention-to-treat analysis found higher HIV care coverage (PRR: 1.06, 95% CI: 1.01 to 1.10, $p = 0.011$) and ART coverage (PRR: 1.05, 95% CI: 1.01 to 1.10, $p = 0.028$) among HIV–positive participants in the intervention compared with the control arm. Male circumcision coverage among all men (PRR: 1.05, 95% CI: 0.96 to 1.14, $p = 0.31$) and HIV viral suppression among HIV–positive participants (PRR: 1.04, 95% CI: 0.98 to 1.12, $p = 0.20$) were higher in the intervention arm, but differences were not statistically significant. No differences were seen in secondary outcomes. Study limitations include reliance on self-report for programmatic outcomes and substantial contamination which may have diluted estimates of effect.

## Conclusions

A novel community health worker intervention improved HIV care and ART coverage in an HIV hyperendemic setting but did not clearly improve male circumcision coverage or HIV viral suppression. This community-based, implementation strategy may be a useful component in some settings for HIV epidemic control.

## Trial registration

ClinicalTrials.gov NCT02556957.

### Author summary

#### Why was this study done?

- HIV remains a major global cause of morbidity and mortality.
- Some communities have very high levels of HIV infection.
- These hyperendemic communities may benefit from new strategies to better implement proven HIV treatment and prevention services.
- A client-centered and mobile phone application-supported counseling approach used by community health workers may be a new strategy to improve the uptake of HIV treatment and prevention services in these settings.

**What did the researchers do and find?**

- We conducted a cluster randomized controlled trial to evaluate the effectiveness of a new community health worker strategy to improve HIV services uptake in a hyperendemic community in Uganda.

- These community health workers were deployed in this community for about 3 years. We compared outcomes in clusters randomized to community health workers with clusters randomized to not having community health workers.

- We found that residents in areas, which had the community health workers, were more likely to be in HIV care and to be on antiretroviral therapy. We found no significant changes in HIV viral suppression or male circumcision coverage.

**What do these findings mean?**

- These findings suggest that this novel community health worker approach may be a useful strategy to help people living with HIV engage in care and get effective treatment.

- This type of intervention may contribute to controlling HIV in settings with high HIV burden.

## Introduction

Combination HIV interventions (CHI), which include HIV Testing Services (HTS), antiretroviral therapy (ART), and voluntary medical male circumcision, have led to declines in HIV incidence in certain populations [1,2]. However, there is limited evidence on effective community-based implementation strategies for achieving high and sustained CHI coverage, particularly in HIV hyperendemic settings [3]. Pragmatic evaluations of comprehensive CHI strategies are needed to inform HIV programs and policy.

HIV hyperendemic settings are geographically defined areas of high HIV transmission and prevalence [4]. Rigorous evidence on CHI implementation strategies for these settings is of particular importance as they are a priority focus of the global HIV response [5–7]. Many fishing communities on Lake Victoria in East Africa are HIV hyperendemic settings characterized by high rates of mobility, condomless sex, multiple sex partners, sex work, alcohol misuse, and inadequate health services [8,9].

Community health workers (CHWs) [10–14], using theory-based frameworks, motivational interviewing counseling techniques, and leveraging mobile health tools to identify and address barriers, may represent a strategy to promote CHI engagement and retention [15,16]. We conducted a pragmatic, cluster-randomized trial called mHealth Lakefolk Actively Keeping Engaged (mLAKE) in an HIV hyperendemic fishing community (baseline HIV prevalence: approximately 38%; baseline ART coverage: approximately 67%; and HIV viral suppression: approximately 66% among HIV–positive residents) in Rakai, Uganda, to assess the impact on HIV service coverage and virologic suppression of a novel motivational interviewing-informed, theory-based, and mobile health-supported community health worker intervention called "Health Scouts."

## Methods

### Study design

The study design and protocol has been previously described in detail [17]. In brief (also see S1 CONSORT Checklist), the study was a pragmatic, parallel, cluster randomized, controlled trial with an allocation ratio of 1:1. The cluster randomized design was chosen to optimize external validity and rigor for this community-based intervention and to minimize contamination threats, which would have been substantial with household- or individual-level randomization; quasi-experimental designs were deemed less rigorous. The study design was pragmatically oriented as our primary goal was to understand Health Scout intervention impact under usual care conditions [18].

### Ethics statement

Ethical approval was obtained from the Johns Hopkins University School of Medicine Institutional Review Board, the Research and Ethics Committee of the Uganda Virus Research Institute, and the Uganda National Council for Science and Technology. All participants provided oral and written informed consent and assent as appropriate.

### Participants

**Study setting.**   The study setting was a single fishing community (area: approximately 2 km$^2$; adolescent/adult population: approximately 4,000 to 5,000) on Lake Victoria in Rakai District of south-central Uganda. Since 2011, the Rakai Health Sciences Program (RHSP) has been the primary provider of CHI services in this fishing community. These services include an HIV clinic, community-based HIV testing, and mobile male circumcision camps. Oral pre-exposure prophylaxis (PrEP) became available to community residents in 2017. ART was initiated at the time of diagnosis (i.e., universal test and treat) throughout the study period. The HIV prevalence in this community was approximately 38% at the start of the study.

**Eligibility criteria for clusters and participants.**   This fishing community, the largest in the region, was divided into 40 clusters. Contiguous clusters were defined with consideration of implementation logistics, minimizing contamination, and geographic features such as roads, buildings, and the lake shoreline. Each cluster was designed to contain roughly the same number of eligible households (approximately 60) and total eligible participants (approximately 107). All resident persons ≥15 years of age in intervention clusters were eligible to receive the Health Scouts intervention. Baseline characteristics and intervention impact was evaluated through the Rakai Community Cohort Study (RCCS) which included all persons aged 15 to 49 years who were resident in a household within the community for at least 1 month or resident for less than 1 month but with intent to stay. RCCS participation was not a requirement for receiving the Health Scout intervention.

**Participant recruitment.**   Recruitment began with a community-wide sensitization in which community leaders and residents were informed about the trial through a series of meetings and public announcements. Health Scouts began visiting participants or "clients" in their households on September 21, 2015. In-migrating participants could be recruited at any time Health Scouts encountered them residing in an intervention cluster during the study period. Participants could also out-migrate or decline participation at any time. Health Scouts would not continue to follow participants if they moved residence from an intervention cluster to a control cluster. The recruitment for the RCCS followed usual procedures [8].

### Health scout intervention

**Conceptualization.**   The Health Scout intervention and initial implementation have previously been described in detail [17,19]. A situated Information, Motivation, and Behavioral Skills (sIMB) theory-based conceptual framework (Fig A in S1 Text) was developed, wherein the Health Scout counseling intervention was designed to promote relevant tailored information, motivation, and behavioral skills to improve client engagement in HIV treatment and prevention services [20,21]. The general approach to the design and implementation of the Health Scouts intervention was pragmatically oriented [18]. That is, while a framework for Health Scouts recruitment, training, tasks, quality assurance, and monitoring was developed, the intervention was designed with flexibility to adapt to any needs, constraints, and secular changes encountered during implementation. Core components of the intervention included (1) CHW (Health Scout)-based delivery; (2) counseling using motivational interviewing-informed strategies; (3) household-based delivery; and (4) mobile health-supported.

**Health Scout phone application.**   The Health Scout Android-based application (Fig B in S1 Text) functioned as a decision and counseling support mobile health tool to guide Health Scouts through counseling sessions (emocha Mobile Health Inc., Baltimore, Maryland, United States of America) [17]. Using simple forms and an sIMB-guided algorithm (Fig C and Table A in S1 Text), the application guided the Health Scout sequentially through 3 steps: (1) Household members were screened for age eligibility; (2) eligible residents were asked a series of triage screening questions, e.g., age, gender, HIV status, to determine which counseling modules should be activated; and (3) tailored modules with motivational interviewing-informed prompts and counseling messages were provided for the Health Scout to review with the client, focusing on areas where the client could most benefit given their current HIV service needs, information, motivation, and behavioral skills, tailored to individual and community contexts. A CHI approach was incorporated, with 10 counseling modules targeting both HIV–positive and HIV–negative residents (Table A in S1 Text).

**Implementation.**   Community mobilization included informational meetings with community leaders and public drama shows depicting proposed Health Scout activities [19]. Ten Health Scouts were initially elected and recruited from the community and underwent residential- and field-based trainings on application and smartphone use, confidentiality, disclosure, HIV-related knowledge, cluster boundaries, and avoiding contamination [19]. Motivational interviewing skills, a communication strategy that emphasizes a nonjudgmental and nonconfrontational approach to behavior change, were integrated into training with core competencies evaluated [19,22,23]. Health Scouts were each initially assigned 2 clusters, which together contained an average of about 215 eligible clients at baseline. Health Scouts attempted to visit all clients within their clusters within 3 months of study initiation and follow up each client within their cluster once every 3 months for approximately 3 years. During each visit, Health Scouts would systematically approach clients at their households and, after receiving oral consent, counsel clients with assistance from systematic prompts from the smartphone application. All initial visits occurred at the clients' residence, but follow-up visits could occur at an alternate place, e.g., work location, based on clients' preferences. Health Scouts were provided a phone, work supplies, and compensation for their work (approximately 60USD a month).

### Control arm

As Health Scouts resided in the community, control arm residents could interact with Health Scouts as they would in routine encounters, but Health Scouts were instructed not to conduct mobile health-supported counseling home visits with residents in the control clusters.

Household residents in both study arms had access to the standard of care HIV prevention and treatment services, which included free HTS, ART, and condoms at a community-based HIV clinic; peer services for patients on ART [24]; access to male circumcision through periodic mobile male circumcision camps; and, beginning in 2017, oral PrEP.

## Outcomes

The trial had 3 primary programmatic outcomes (self-reported linkage to HIV care, ART coverage, and male circumcision coverage) and 1 primary biologic outcome (HIV viral suppression, defined as <400 copies/mL, among HIV–positive participants; see Table B in S1 Text for detailed outcome definitions). Secondary outcomes included HTS coverage, HIV incidence, and consistent condom use. The study was nested within the RCCS, which was used to assess baseline characteristics and all study outcomes through 3 community-wide, cross-sectional surveys. RCCS surveys were conducted at baseline (approximately 4 months prior to intervention start), at the study midpoint (approximately 15 months after intervention start) and at the study end-point (approximately 39 months of total follow-up). The RCCS is an open, population-based cohort focused on HIV surveillance among persons 15 to 49 years of age [25]. Since 2011, the RCCS has surveyed all age-eligible individuals, regardless of HIV status, in this community approximately every 18 months [8]. At each survey, the RCCS first conducts a census of households with GPS coordinates recorded, and all resident household members are enumerated by sex, age, and duration of residence. These GPS coordinates were used to classify participants into specific study clusters. After census, the RCCS conducts interviews with consenting participants to assess demographics, sexual behaviors, and HIV service uptake. Blood samples are collected for HIV viral load assays. Following the interview, free HIV testing services with results immediately returned are offered to consenting participants.

## Sample size

The sample size for this trial was fixed at 40 clusters (20 intervention, 20 control, approximately 107 eligible participants in each cluster); therefore, we estimated whether the trial had power to detect relevant and realistic differences in primary study outcomes, i.e., minimum detectable differences between arms, based on our prior experience with CHW interventions and mobile health tools in Uganda [24,26–28]. Detailed power calculation inputs and results were previously described [17]. In summary, the trial was estimated to have the power to detect minimum difference prevalence risk ratios with 80% power of ≥15% differentials in HIV care coverage, ≥20% in ART coverage, ≥41% in viral suppression, and ≥23% in male circumcision coverage.

## Randomization

Given the fixed sample size and desire for baseline comparability of important characteristics, we used restricted randomization to 1:1 allocated intervention and control arms [26]. First, we defined a priori acceptable differences in baseline cluster level characteristics [17]. Second, we conducted independent, computer-generated randomizations with those not satisfying comparability criteria removed. Subsequently, 1 randomization sequence was randomly drawn for the allocation of intervention. The final assignment of clusters to intervention and control arms was done in a public coin flip ceremony witnessed by study staff and the Health Scouts. By the nature of the intervention, Health Scouts and participants were not blinded to the intervention assignment. The study team and primary analyst were blinded to the intervention assignment during analyses.

## Statistical analysis

Individual-level data for study outcomes were used to estimate the efficacy of the intervention. The primary analysis was by intent-to-treat and used log-linear binomial models to estimate prevalence risk ratios for each outcome comparing the intervention versus the control arm. A generalized estimating equation with an exchangeable correlation structure was used to account for within-cluster correlations. Separate analyses were conducted for each RCCS follow-up survey to assess the shorter (mid-study survey) and longer-term (end-study survey) effects of the intervention. A serial cross-sectional analysis approach was chosen a priori, as high loss to follow-up was anticipated in this mobile population, and our main interest was outcomes across the entire population present at the time of survey and not just participants present at baseline. An intention-to-treat analysis adjusting for age and sex was also performed. Finally, an as-treated analysis was conducted with exposure defined as self-reported receipt of any Health Scout counseling (exposure data was also collected through the RCCS survey).

# Results

A baseline survey ($n$ = 2,148) was conducted from April 7, 2015, to July 14, 2015, (survey interval midpoint: May 5, 2015). As shown in Table 1, baseline characteristics were generally well balanced between study arms except control arm participants were less likely to be married and more likely to have a secondary/tertiary education (Table C in S1 Text). Beginning on September 21, 2015, the Health Scout intervention was implemented in the intervention clusters and continued through December 19, 2018. A mid-study survey ($n$ = 2,533) was conducted from December 5, 2016 to February 13, 2017 (survey interval midpoint: January 12, 2017). An end-study survey ($n$ = 1,903) was conducted from September 3, 2018, to December 19, 2018 (survey interval midpoint: October 5, 2018). At the mid- and end-of-study surveys, participant characteristics remained mostly well balanced between study arms (Table C in S1 Text).

As shown in Fig 1, in this highly mobile study population [29], large numbers of participants moved in and out of the community between surveys, with smaller numbers of participants moving between intervention and control clusters between surveys. While the number of study participants varied from survey to survey, the overall survey response rates of residents who were eligible and present at the time of survey was consistent for the baseline, mid-study, and end-study surveys, i.e., 99.4%, 99.0%, and 99.5% respectively.

## Implementation measures

Ten Health Scouts were initially trained and deployed. Over the study period, 3 Health Scouts were eventually relieved for not following study protocols despite retraining, and 2 Health Scouts left for other opportunities. Only 3 new Health Scouts were hired as replacements, as experience showed that 8 Health Scouts were sufficient to accomplish the workload.

From study initiation to the start of the end-study survey period, Health Scouts completed a total of 11,221 counseling sessions among 2,532 unique residents. The median number of sessions per client was 4 (IQR: 2 to 7); the mean number of sessions per client was 4.43 (range: 1 to 18). A total of 614 clients received only 1 counseling session.

At the mid-study survey, participants' self-reported history of having ever been visited and counseled by a Health Scout (Table D in S1 Text), i.e., exposed, was higher in the intervention (31%) compared with that in the control (15%) arm and varied among subgroups, e.g., with both arms combined, the exposure was higher among those living with HIV (30%) compared with those who were not (18%) and higher among women (28%) compared with among men (18%). By the end-study survey, the self-reported history of having ever been visited and

**Table 1. Baseline, individual-level characteristics by study arm.**

| Characteristics | Intervention | Control |
|---|---|---|
| N | 1054 | 1094 |
| Mean (SD) age (years) | 30.6 (7.71) | 30.1 (7.87) |
| Age years | | |
| 15–24 | 253 (24.0) | 296 (27.1) |
| 25–34 | 471 (44.7) | 454 (41.5) |
| 35–49 | 330 (31.3) | 344 (31.4) |
| Sex | | |
| Female | 535 (50.8) | 511 (46.7) |
| Male | 519 (49.2) | 583 (53.3) |
| Marital status | | |
| Married | 664 (63.0) | 613 (56.0) |
| Never married | 122 (11.6) | 182 (16.6) |
| Previously married | 268 (25.4) | 299 (27.3) |
| Educational status | | |
| None | 102 (9.7) | 80 (7.3) |
| Primary | 770 (73.1) | 762 (69.7) |
| Secondary/tertiary | 182 (17.3) | 252 (23.0) |
| Religion | | |
| Christian/non-Muslim | 888 (84.3) | 894 (81.7) |
| Muslim | 166 (15.7) | 200 (18.3) |
| Occupation | | |
| Agriculture/housework | 191 (18.1) | 177 (16.2) |
| Bar/restaurant | 112 (10.6) | 109 (10.0) |
| Fishing | 304 (28.8) | 303 (27.7) |
| Trade/shopkeeper | 228 (21.6) | 223 (20.4) |
| Other | 219 (20.8) | 282 (25.8) |
| Male circumcision (among men) | | |
| Yes | 292 (56.3) | 353 (60.5) |
| No | 227 (43.7) | 230 (39.5) |
| HIV serostatus | | |
| Positive | 400 (38.1) | 386 (35.3) |
| Negative | 650 (61.9) | 708 (64.7) |
| HIV serostatus awareness | | |
| Yes | 1006 (95.4) | 1043 (95.3) |
| No | 5 (0.5) | 2 (0.2) |
| Never Tested | 43 (4.1) | 49 (4.5) |
| In HIV care (among all HIV+) | | |
| Yes | 309 (77.3) | 292 (75.6) |
| No | 91 (22.8) | 94 (24.4) |
| On ART (among all HIV+) | | |
| Yes | 271 (67.8) | 259 (67.1) |
| No | 129 (32.3) | 127 (32.9) |
| HIV viral suppression (among all HIV+) | | |
| Yes | 270 (67.5) | 248 (64.2) |
| No | 130 (32.5) | 138 (35.8) |

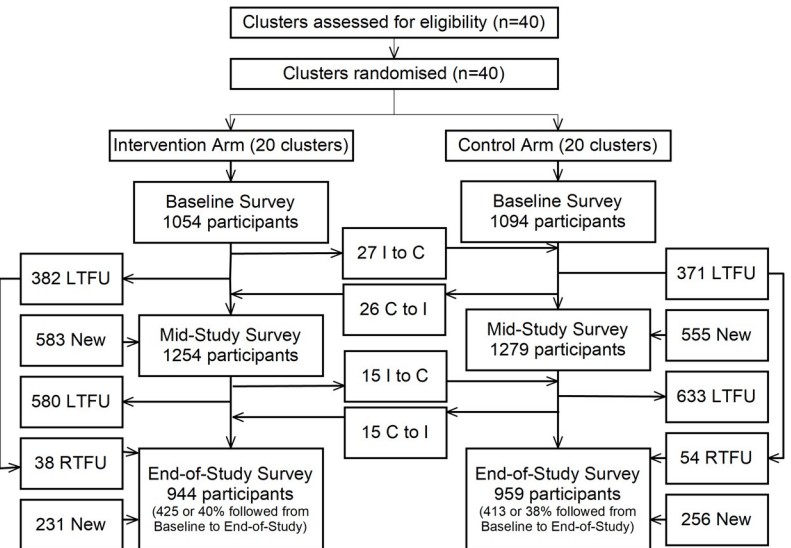

**Fig 1. Study flow diagram.** Three surveys were conducted (baseline, mid-study, and end-study). Participants could be LTFU from 1 survey to the next, RTFU from the baseline to end-study survey, be new to a survey, or move between I and C clusters between surveys. C, control; I, intervention; LTFU, lost-to-follow-up; RTFU, return-to-follow-up.

counseled by a Health Scout was higher in the intervention (38%) compared with that in the control (23%) arm, among those living with HIV (42%) compared with those who were not infected (24%) and among women (35%) compared with among men (27%).

## Intention-to-treat analyses of primary outcomes

At the mid-study survey, there were 913 participants who were HIV–positive (36.0%). As shown in Table 2, HIV care coverage, ART coverage, and HIV viral suppression outcomes among all HIV–positive participants did not differ between study arms at the mid-study survey. At the mid-study survey, among 1,396 men, male circumcision coverage did not differ by study arms.

At the end-study survey, there were 679 participants who were HIV–positive (35.7%). In contrast to the mid-study survey, HIV care coverage and ART coverage outcomes were higher in the intervention arm compared with those in the control arm (Table 2). HIV viral

**Table 2. Intention-to-treat results for primary outcomes comparing study arms.** [a]

| Outcome | Mid-study survey | | | | | End-study survey | | | | |
|---|---|---|---|---|---|---|---|---|---|---|
| | N | Intervention | Control | PRR (95% CI) | p-Value | N | Intervention | Control | PRR (95% CI) | p-Value |
| HIV care coverage | 913 | 393/466 (84.3%) | 381/447 (85.2%) | 0.99 (0.92–1.06) | 0.79 | 679 | 327/350 (93.4%) | 290/329 (88.2%) | 1.06 (1.01–1.10) | 0.011 |
| ART coverage | 913 | 382/466 (82.0%) | 362/447 (81.0%) | 1.01 (0.94–1.09) | 0.70 | 679 | 323/350 (92.3%) | 288/329 (87.5%) | 1.05 (1.01–1.10) | 0.028 |
| HIV viral suppression | 901 | 363/460 (78.9%) | 336/441 (76.4%) | 1.03 (0.96–1.11) | 0.38 | 656 | 300/338 (88.8%) | 269/318 (84.6%) | 1.04 (0.98–1.12) | 0.20 |
| Male circumcision coverage | 1396 | 444/685 (64.8%) | 474/711 (66.7%) | 0.96 (0.87–1.07) | 0.47 | 1032 | 340/494 (68.9%) | 354/538 (65.8%) | 1.05 (0.96–1.14) | 0.31 |

[a]ART coverage and HIV viral suppression are among all HIV–positive participants. Male circumcision is among men only.

ART, antiretroviral therapy; PRR, prevalence risk ratios.

**Table 3. Intention-to-treat results for secondary outcomes comparing study arms.**

| Outcome | Mid-study survey | | | | | End-study survey | | | | |
|---|---|---|---|---|---|---|---|---|---|---|
| | N | Intervention | Control | PRR (95% CI) | p-Value | N | Intervention | Control | PRR (95% CI) | p-Value |
| HIV testing coverage | 2533 | 1184/1254 (94.4%) | 1225/1279 (95.8%) | 0.99 (0.96–1.00) | 0.24 | 1902 | 923/944 (97.8%) | 938/959 (97.9%) | 1.00 (0.98–1.01) | 0.81 |
| Consistent condom use | 1353 | 153/668 (20.5%) | 187/685 (27.3%) | 0.84 (0.69–1.03) | 0.10 | 928 | 89/452 (19.7%) | 96/476 (20.2%) | 0.98 (0.78–1.24) | 0.88 |
| HIV incidence[a] | - | - | - | - | - | | 17 cases/1188 py (rate = 0.0143) | 12 cases/1353 py (rate = 0.0089) | IRR = 1.19 (0.46–3.09) | 0.72 |

[a]HIV incidence was calculated at end-study only as an IRR.

IRR, incidence rate ratio; PRR, prevalence risk ratios; py, person-years.

suppression among HIV–positive participants and male circumcision among all men were numerically higher in the intervention compared with those in the control arm, but the difference was not statistically significant.

## Ancillary analyses

An intention-to-treat analysis adjusting for age and sex (Table E in S1 Text) was largely consistent with the unadjusted analysis, but notable differences included significant improvement in HIV care and ART coverage in the intervention arm compared with those in the control arm at the mid-study survey rather than just the end-study survey. Analysis of secondary outcomes did not find any significant differences in HTS coverage, consistent condom use, or HIV incidence at mid-study or end-study (Table 3). Finally, a not prespecified, unadjusted, and age and sex–adjusted as-treated analysis found that self-reported exposure to the Health Scout intervention was associated with increased HTS, HIV care, and ART coverage at both mid-study and end-study and increased HIV viral suppression at mid-study compared with unexposed participants (Table F in S1 Text).

## Discussion

In this cluster randomized trial, a motivational-interviewing informed, theory-based, mobile health-supported CHW intervention moderately improved HIV care and ART coverage among residents of an HIV hyperendemic fishing community in Uganda. Viral suppression and male circumcision coverage were not significantly improved in the intention-to-treat analysis, but the direction of effect consistently favored the intervention arm across all outcomes. Adjusted and an as-treated analyses were supportive of the primary intention-to-treat analysis. Prior CHW-based HIV interventions in Africa have incorporated mobile health tools [28,30], motivational interviewing [31,32], and the sIMB framework [33], but to our knowledge, this is the first report of an effective intervention combining all of these elements.

While the absolute difference in outcomes was modest, e.g., approximately 5% difference in ART coverage, these differences may have meaningful public health impact as small improvements in settings with already high levels of coverage and viral suppression could be needed for HIV epidemic control [34]. Comprehensive, community-based interventions such as the Health Scouts may contribute to the incremental gains in HIV care outcomes needed to reduce HIV incidence in similar settings, particularly if the intervention facilitates the recruitment of hard-to-reach individuals. As CHWs are often present in HIV-affected communities, adopting the Health Scouts approach into ongoing programs could offer added benefits in HIV treatment and prevention.

This study had a number of limitations. According to self-report, a substantial number of participants residing in control clusters at the time of survey had been counseled by a Health Scout, indicating contamination of the control arm. Participant flow (Fig 1) from intervention to control arms and vice versa appeared modest; however, these data only captured residence at the time of survey, and participants may move residence more than once during inter-survey intervals, e.g., many participants rely on short-term rentals or transient sleeping under boats. Participants were also classified in intervention and control arms based upon household Global Positioning System (GPS) data collected using handheld GPS devices, which can be imprecise, potentially resulting in misclassification. Some cluster boundaries were also physically altered by shoreline erosion and demolition and construction of houses, potentially leading to contamination and/or misclassification. However, contamination and nondifferential misclassification would have likely biased results toward the null, suggesting that the Health Scouts effect is likely greater than observed. Future studies may consider having entire communities randomized to decrease contamination risks.

The study population was highly mobile, as evidenced by the high rates of lost-to-follow-up, return-to-follow-up, and movement between study arms. However, the overall effect of this mobility would most likely also be a dilution of measured impact. Based on self-report, the intervention reach in this study was moderate with only 38% of residents reporting receiving counseling by the Health Scouts in the intervention arm by study end. Of note, if there had been no contamination and 100% reach in the intervention clusters, the overall reach in all 40 clusters would be maximized at 50%, and this study had 30% overall community reach. Regardless, the moderate reach in intervention clusters may have reduced community-wide impact, and additional evaluations of barriers and facilitators to intervention implementation are planned. Study outcomes, except for viral suppression, were based on self-report, as were measures of Health Scouts exposure. Self-report has known limitations, particularly social desirability bias, but we have previously validated self-report of ART use and male circumcision in this setting [35,36].

Of note, no significant differences in outcomes were seen at the mid-study survey in the primary intention-to-treat analysis. The intervention reach may have been insufficient at that time to produce detectable impacts on outcomes. Health Scouts may also have needed time to develop trusting relationships with clients to better affect behavior change. Despite contamination between the groups increasing between mid-study and end-study, effect measures did reach statistical significance at the end-study. Further evaluations of the long-term implementation and sustainability of this approach could be informative.

In summary, a novel, theory-based community health worker intervention using motivational interviewing techniques and a mobile health tool was associated with improved HIV service engagement in a highly mobile HIV hyperendemic setting. This community-based intervention may be a useful component of a comprehensive response aiming for 95-95-95 and HIV epidemic control [37]. Future mixed-methods evaluations of this implementation strategy will be informative to better understand important issues around implementation, replicability, costs, and sustainability.

## Supporting information

**S1 CONSORT Checklist. CONSORT checklist for reporting a cluster randomized trial.**
(DOCX)

**S1 Text. Supplementary figures and tables.**
(DOCX)

## Acknowledgments

We thank the Health Scouts, the study participants, and the staff of the Rakai Health Sciences Program.

## Author Contributions

**Conceptualization:** Larry W. Chang, Ismail Mbabali, Heidi Hutton, K. Rivet Amico, Jeremiah Mulamba, Caitlin E. Kennedy, Robert C. Bollinger, Thomas C. Quinn, Steven J. Reynolds, Ronald H. Gray, Maria J. Wawer, Gertrude Nakigozi.

**Data curation:** Aggrey Anok, Joseph Ssekasanvu.

**Formal analysis:** Xiangrong Kong, Aggrey Anok, Joseph Ssekasanvu.

**Funding acquisition:** Larry W. Chang, Robert C. Bollinger.

**Investigation:** Heidi Hutton, K. Rivet Amico, Caitlin E. Kennedy.

**Methodology:** Xiangrong Kong.

**Project administration:** Larry W. Chang, Ismail Mbabali, Heidi Hutton, K. Rivet Amico, Jeremiah Mulamba, Aggrey Anok, Amanda Long, Alvin G. Thomas, Kristin Thomas, Eva Bugos, Rose Pollard, Kimiko van Wickle, Fred Nalugoda, David Serwadda, Thomas C. Quinn, Steven J. Reynolds, Gertrude Nakigozi.

**Supervision:** Larry W. Chang, Ismail Mbabali, Amanda Long, Alvin G. Thomas, Kristin Thomas, Eva Bugos, Rose Pollard, Kimiko van Wickle, Caitlin E. Kennedy, Fred Nalugoda, David Serwadda, Ronald H. Gray, Maria J. Wawer, Gertrude Nakigozi.

**Writing – original draft:** Larry W. Chang, Heidi Hutton.

**Writing – review & editing:** Ismail Mbabali, K. Rivet Amico, Xiangrong Kong, Jeremiah Mulamba, Aggrey Anok, Joseph Ssekasanvu, Amanda Long, Alvin G. Thomas, Kristin Thomas, Eva Bugos, Rose Pollard, Kimiko van Wickle, Caitlin E. Kennedy, Fred Nalugoda, David Serwadda, Robert C. Bollinger, Thomas C. Quinn, Steven J. Reynolds, Ronald H. Gray, Maria J. Wawer, Gertrude Nakigozi.

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
