## [Editor Report · Decision Letter 0]

27 Jul 2020

Dear Dr Chang, 

Thank you for submitting your manuscript entitled "Novel Community Health Worker Strategy for HIV Service Engagement in a Hyperendemic Community in Rakai, Uganda: A pragmatic, cluster-randomized trial" for consideration by PLOS Medicine.

Your manuscript has now been evaluated by the PLOS Medicine editorial staff, and I am writing to let you know that we would like to send your submission out for external assessment.

Kind regards,

Richard Turner, PhD

Senior editor, PLOS Medicine

rturner@plos.org

---

## [Decision Letter · Decision Letter 1]

21 Aug 2020

Dear Dr. Chang,

Thank you very much for submitting your manuscript "Novel Community Health Worker Strategy for HIV Service Engagement in a Hyperendemic Community in Rakai, Uganda: A pragmatic, cluster-randomized trial" (PMEDICINE-D-20-03569R1) for consideration at PLOS Medicine. 

Your paper was evaluated by the editors and sent to independent reviewers, including a statistical reviewer. The reviews are appended at the bottom of this email and any accompanying reviewer attachments can be seen via the link below:

[LINK]

In light of these reviews, we will not be able to accept the manuscript for publication in the journal in its current form, but we would like to invite you to submit a revised version that addresses the reviewers' and editors' comments fully. You will appreciate that we cannot make a decision about publication until we have seen the revised manuscript and your response, and we expect to seek re-review by one or more of the reviewers. 

We hope to receive your revised manuscript by Sep 11 2020 11:59PM. Please email us (plosmedicine@plos.org) if you have any questions or concerns.

Please let me know if you have any questions. Otherwise, we look forward to receiving your revised manuscript in due course. 

Sincerely,

Richard Turner, PhD

rturner@plos.org

Please adapt the data statement to include details for a non-author contact for inquiries. 

Please quote aggregate demographic details for study participants in the abstract. 

Please list the study's primary and secondary endpoints early in the "Methods and findings" subsection of your abstract. 

In the abstract and elsewhere in the paper, please add p values alongside 95% CI, where available. 

Please add a new final sentence to the "Methods and findings" subsection of your abstract, quoting 2-3 of the study's main limitations. 

After the abstract, we will need to ask you to add a new and accessible "author summary" section in non-identical prose. You may find it helpful to consult one or two recent research articles published in PLOS Medicine to get a sense of the preferred style. 

At the start of the "Discussion" section in the main text, please begin the first paragraph with "In this cluster-randomized trial ..." or similar. 

Please remove the information on funding at the end of the main text. In the event of publication, this information will appear in the article metadata, via entries in the submission form. 

Please adapt the reference call-outs to remove spaces within the square brackets (e.g., "... health services [8,9].").

In the reference list, please convert italics and boldface to plain text. Where appropriate, 6 author names should be listed, followed by "et al.". 

Please add a completed CONSORT checklist as a supplementary document, referred to early in the methods section (e.g., "See S1_CONSORT_Checklist"). In the checklist, individual items should be referred to by section (e.g., "Methods") and paragraph number rather than by line or page numbers, as the latter generally change in the event of publication. 

Comments from the reviewers:

*** Reviewer #1: 

This article, describes the finding from a pragmatic randomized trial on novel Community Health Worker intervention, called "Health Scouts" for HIV Service Engagement defined as (adherence to ART therapy, viral suppression and male circumcision) in a Hyperendemic Community in Rakai,Uganda. One of the strengths of the study is that is uses a Theory based intervention grounded in a Motivational Interviewing Framework. The use of pragmatic, cluster- randomized trial also provides a rigorous evidence on the effectiveness of CHW interventions for people with HIV in improving health outcomes. The use of a mobile application in a hypendemic, low resource setting is also useful to the field. However there are a number of limitations that need to be addressed prior to publication. The article could be strengthened by stating clearly up front the recruitment procedures, clear eligibility for the study population and more in-depth information about the intervention itself. Furthermore there appears to be great loss to follow up on the intervention from baseline to mid study and end study period thus limiting the reliability and validity of the study findings. 

Abstract

* How is self-care defined?

Introduction

* The introduction begins by describing combination HIV Interventions (CHI) of HIV testing, male circumcision, and antiretroviral therapy. How does the CHW intervention and/or mobile application link to the CHI.

* HIV prevalence is 38%--what is the level of viral suppression? Is ART available? 

Methods

Study design: what is usual care that the Health Scout intervention is compared to?

Participant eligibility criteria: If the study is about HIV care coverage and ART therapy—why is the eligibility for this study limited to just people HIV >15 rather than all resident. Or state that this is sub-study of the larger study. 

Participant recruitment: page 7 Health Scouts started visiting "clients". How were clients contacted? Describe the community mobilization techniques? Did CHWs going randomly from house systematically and inform clients about the program and see if someone wanted to participate.?

Phone application Intervention: How many modules? what was the protocol for delivering modules? 

Implementation: How many clusters were each CHW assigned? So Health Scout CHWs were in all 40 clusters? Mobile clusters were only in a few? How many clients in each cluster; how often follow up in 3 month follow-up (weekly?) at least once? 

Control arm: what was the standard of care? 

Outcomes: If person tests positive are they informed of the results? How is HIV self-care coverage? How is MC coverage measured? How is ART measured? 

Sample size: provide 2-3 sentences about the power calculation and estimated size of each population in the clusters

Statistical Analysis: The analysis and results could be strengthened if authors examined the sample that had baseline and follow up data either mid point or end survey. 

Results-Table

Is there an attrition analysis? According to tables there is significant lost to followup between baseline, mid point and end survey. How many people had completed data ie. at least one baseline and follow up.

How did you calculate the response rates for follow time points? Given the samples size at mid year and end study more than 50% of the sample 425 (baseline to end survey) and 413 for controls. AS new participants were recruited at mid year -were they matched to 

Implementation measures: The average of 4 visits over 3 years seems quite low over 3 years. What was the average time in the intervention? Is this just for people with HIV? nearly 25% received only 1 session--why such a big LTF? What was the protocol? 

Only 38% of the intervention group received the Health Scout intervention with the mobile app. This seems low for an intervention design. 

For those who are not HIV infected what were their outcomes of interest? It is not clear why they are included in this study. 

Discussion

Is the intervention sustainable? It seems there was high turnover among Health SCOUTS from 10 to approximately 5? 

LImitations: The authors duly note that contamination could be reasons for null effects between intervention and control groups. Was supervision provided to Health Scouts to try to limit contamination. What further recommendation for future study designs and would the authors recommended? Step Wedge?

*** Reviewer #2: 

PMEDICINE-D-20-03569R1

This pragmatic cluster RCT examines the effectiveness of a community-based motivational interviewing m-health intervention embedded within the Rakai cohort. The study design and careful plan are strengths. However, there are some loose ends that would be worth clarifying further. In particular, more discussion of the pilot data, contamination, LTFU, and intervention would enhance the value of this study.

MAJOR COMMENTS

(1) Pilot data. Given that there was no difference between the two arms, it would be helpful to know about the pilot data that informed the sample size calculation and study design. However, I could not find published pilot data demonstrating the effectiveness (beyond feasibility) of this specific intervention in the protocol manuscript or this submitted paper. Could this data be highlighted?

(2) Contamination. The extent of contamination in Figure 1 seems rather modest. While I agree this would bias towards the null, the sample size calculations accounted for 10-50% contamination. A more thorough discussion of contamination and mention of this in the abstract would be useful for the reader. 

(3) Rakai cohort LTFU and migration. The study was nested within the Rakai open cohort, making this one of the most extensively researched populations in the world. Given this context, it is surprising that migration and LTFU were such unanticipated problems. Did something change with the cohort over time or is there some other explanation for these differences? 

(4) Health Scouts. More details on these individuals either in the text or a supplement would be useful. Based on looking at the protocol, they had no experience with motivational interviewing and were tasked with implementing a substantial public health task with minimal mentorship. How was "basic competency with assigned tasks" evaluated in the field? What support was provided and how were they compensated?

(5) M-health data. Was the back-end data from app use also used to see levels of use? This implementation data might be able to what extent scouts were implementing the standard intervention.

MINOR COMMENTS

(1) Abstract, results section. Contamination. Given that contamination was an important part of this study, including some basic description of the scope of contamination in the abstract would be important. 

(2) Abstract, results section, sentence starting with "At end-study, self-reported.." I believe that the counselling was the intervention itself, correct? Would consider revising this sentence to focus on exposure to the intervention in the two study arms. 

(3) Secondary outcomes. It would also be useful to see the pre-specified secondary outcomes of interest (condomless sex, viral load, testing coverage, etc).

(4) Per protocol analysis. The protocol published in Trials does not specify a per-protocol analysis. Was this added later? It's a nice addition to the paper, but would be appropriate to note that this was not pre-specified (if this was the case).

(5) Figure 2S. Its great to have some screen shots, but would be good to have a larger high-resolution version of this content. 

*** Reviewer #3: 

Alex McConnachie, Statistical Review

Chang et al present the results of a cluster randomised trial of a community health worker intervention to improve HIV service engagement in a hyperendemic setting in Uganda. This review considers the use of statistics in the paper.

Overall, I found the paper to be interesting, and well written. The study background, methods, and results, are described clearly.

The sample size justification is reasonable; studies like this often have a fixed sample size, and the question is whether the study has power to detect a reasonable effect. There is a reference to another paper with details of the sample size calculations, which is fine, but I think in the current paper it would be good to state what minimum effect size the study was powered to detect.

The statistical analysis methods are acceptable, as far as they go. One of the most striking results is the level of contamination between the study arms, with almost a quarter of the control arm reportedly receiving the intervention. Whilst an ITT analysis is perfectly reasonable, I feel the authors should seek to estimate what the true impact of the intervention might be. For example, this could involve a CACE analysis to estimate the effect of the intervention amongst those who received it. This, coupled with the estimated reach of the intervention, could give an estimate of the impact of the intervention over a population. Currently, we have estimates of differences in outcomes between two groups with different levels of access to the intervention, which provide evidence of intervention effects, but do not give good estimates of the magnitude of those effects. What matters, I think, are the differences in outcomes between a population with access to the intervention and the same population without access to the intervention.

*** Reviewer #4: 

This paper reports the primary outcomes of a cluster-randomized trial evaluating the effect of a community health workers intervention (called "Health Scouts") on engagement into HIV care and prevention measured at the population level.

The trial was embedded into a population cohort used to measure the different outcomes at baseline, mid-point and end line.

Implementing and evaluating complex combination interventions are crucial to achieving epidemic control. The results reported in this paper are entirely relevant in this context.

Overall, the Health Scouts intervention could be more described in the paper, the authors referring to a protocol paper published in Trials (https://doi.org/10.1186/s13063-017-2243-6). In particular, it is unclear what are the counselling messages delivered by the scouts. The paper is focused on the 4 primary outcomes of the trial, 3 of them concerning only HIV-positive individuals (HIV care coverage, ART coverage and viral suppression) and one among men (male circumcision). When just reading the paper, it seems strange that all individuals were eligible for the scout intervention as it was unclear that would be the counselling for negative women.

In the protocol paper, it appeared that there are 9 counselling modules: 1 HIV serostatus unknown or no recent HIV test, male or not pregnant female; 2 HIV serostatus unknown or no recent HIV test, pregnant female; 3 Male, not circumcised; 4 HIV+, not in care; 5 HIV+, in care, not on antiretroviral therapy (ART); 6 HIV+, on ART; 7 HIV+, pregnant; 8 Male having condomless sex; 9 Female having condomless sex. However, it is not described whare are the recommendations for each group. For example, are uncircumcised HIV-positive men invited to get circumcised?

According to the protocol paper, there are 4 defined secondary outcomes (HIV testing, HIV incidence, HIV treatment failure and condom use). What are the reasons for not reporting these 4 outcomes in the same paper as the 4 primary outcomes? It would provide a more global view of this global and multi-component intervention.

The paper is relatively silent about what was done by the health scouts, just mentioning 11,221 counselling sessions among 2,532 unique residents. Do the authors have any information/statistic about the content of the sessions? Was data collected in the app about the profile of the beneficiaries? 

Regarding the 4 primary outcomes, they should be more formally defined, with a precise definition of the numerators and the denominators.

It appears in the Results that 3 surveys have been conducted at baseline, mid-point and endline. However, the baseline survey is not mentioned in the methods. Was it also conducted as part of RCCS?

Regarding the analysis of the primary outcomes, it seems that only within-cluster correlations were taken into account, but that the models were not adjusted on sociodemographic characteristics. In the protocol paper, it is written: "If a baseline outcome or demographic variable are significantly different between arms with p-value ≤ 0.1, adjusted regression analysis on the outcome variables will be conducted adjusting for baseline covariates." According to table 1S, at baseline, the two arms differ statistically (p < 0.1) in terms of sex, marital status and educational status. Therefore, could the authors justify the choice of non-adjusting on these variables?

Considering the fact that only a small proportion (38%) of participants self-reported health scouts counselling at endline in the intervention arm and that 23% of participants reported health scouts counselling in the control arm, did the authors consider the possibility of a comparison of the different outcomes between those reporting counselling vs the others, as a complementary analysis to the intent-to-treat analysis?

The authors may consider some rewording of the two paragraphs presenting the intention-to-treat analyses as it is confusing in its current form. The two paragraphs start with the number of HIV-positive participants, suggesting that this subgroup was the denominator of the different outcomes. But this is true only for 3 outcomes (HIV care coverage, ART coverage and viral load). Clearly, male circumcision coverage used a different denominator, not mentioned in the text. Also, could the authors clarify if male circumcision coverage was computed among all men or only among those HIV-negative?

In the discussion, there is the following statement: "The intervention arm in this study was able to achieve these benchmarks" (i.e. 95-95-95 Unaids' targets). However, data to justify this statement are not presented in the paper. Besides, achieving these targets at endline without knowing what was the situation at baseline nor what was achieved in the control arm, doesn't mean anything about the contribution of Heath Scouts in this achievement.

***

[LINK]

---

## [Decision Letter · Decision Letter 2]

1 Nov 2020

Dear Dr. Chang,

Thank you very much for re-submitting your manuscript "Novel Community Health Worker Strategy for HIV Service Engagement in a Hyperendemic Community in Rakai, Uganda: A pragmatic, cluster-randomized trial" (PMEDICINE-D-20-03569R2) for consideration at PLOS Medicine.

I have discussed the paper with editorial colleagues and it was also seen again by 3 reviewers. I am pleased to tell you that, provided the remaining editorial and production issues are fully dealt with, we expect to be able to accept the paper for publication in the journal.

[LINK]

Please let me know if you have any questions. Otherwise, we look forward to receiving the revised manuscript shortly. 

Sincerely,

Richard Turner, PhD

plosmedicine.org

Requests from Editors:

Please confirm that all elements of your paper, including fig B, for example, can be published under a CC BY licence.

You mention a "redacted version" (of the study data) in your data statement. Please explain what is being redacted; it may be that adapting this to "anonymised data" or similar would be preferable as "redacted" may raise questions for readers about what is being obscured. 

Please restructure the abstract so that the 4 primary outcomes are listed (we suggest with the "primary biologic outcome" last) and then the findings on each presented in the same order. 

We suggest quoting the number of participants in the two study arms at baseline, early in the "Methods and findings" subsection of your abstract. 

Please adapt the "Conclusions" subsection of your abstract to note the null findings on two of the primary endpoints. 

Please add bullet points to the author summary; and adapt the final subsection to contain at least 2 points. 

Please adapt the wording of your author summary to substitute "no significant changes", or similar, for "no large changes".

We understand that "hotspots" is not a universally popular term in this context, and ask you to remove it. 

In the (quite brief) introduction and/or discussion sections of your main text, are there similar published interventions that could be mentioned and cited?

At the start of the methods section, please adapt the wording to make it clear that reference 18 is the study protocol. 

Where appropriate, e.g., in the methods section, please substitute "sex" for "gender". 

As discussed in the response-to-referees, please note at the end of the results section that the as-treated/per-protocol analysis was not prespecified. 

Please quote exact p values or "p<0.001" (noting several instances of "p<0.0001" in table G, for example). 

We ask you to revisit the discussion of "direction of effect" early in the discussion section of your main text, so as to avoid any implication that non-significant findings are significant. 

To references 7 and 34, please add URLs and accessed dates, if available. 

Please add an author (or institutional author) name to reference 12. 

To reference 30, please add the first author's initial in place of the first name. 

Noting table C, we understand that the CONSORT guideline discourages baseline statistical comparisons. 

We were unable to find the CONSORT checklist with your resubmission, and ask that you include this with your revision. In the checklist, please ensure that individual items are referred to by section (e.g., "Methods") and paragraph number rather than by line or page numbers, as the latter generally change in the event of publication. 

Comments from Reviewers:

*** Reviewer #1: 

Thank you for your thorough response to the reviewers comments. The response and changes to the manuscript are acceptable and more clearly explain the design, data analysis and limitations and limitation of the findings. 

Reviewer #3: 

Alex McConnachie, Statistical Review

I am happy with the revisions made by the authors in response to my original comments.

The addition of an as-treated analysis does not achieve what I had hoped in terms of an assessment of the potential impact of the intervention for a population, compared to a population without access to the intervention, but on reflection it may not be possible to do a CACE analysis in this setting. The authors do recognise that their intervention effect estimates are likely to be underestimates of the true impact of the intervention. On that basis, I have no additional comments to make.

*** Reviewer #4: 

The authors provided a detailed rebuttal letter and answered to all reviewer's comments. In particular, they added detailed about the intervention, a multivariate-adjusted analysis of the outcomes, the secondary outcomes and an as-treated analysis. The analysis has been clearly improved.

I just have a little comment regarding secondary outcomes: they are presented only in supplementary materials. However, there is enough space in Table 2 to add them to the main article.

***

[LINK]

---

## [Editor Report · Decision Letter 3]

30 Nov 2020

Dear Dr. Chang, 

On behalf of my colleagues and the academic editor, Dr. Joseph D Tucker, I am delighted to inform you that your manuscript entitled "Novel Community Health Worker Strategy for HIV Service Engagement in a Hyperendemic Community in Rakai, Uganda: A pragmatic, cluster-randomized trial" (PMEDICINE-D-20-03569R3) has been accepted for publication in PLOS Medicine. 

PRODUCTION PROCESS

Before publication you will see the copyedited word document (within 5 business days) and a PDF proof shortly after that. The copyeditor will be in touch shortly before sending you the copyedited Word document. We will make some revisions at copyediting stage to conform to our general style, and for clarification. When you receive this version you should check and revise it very carefully, including figures, tables, references, and supporting information, because corrections at the next stage (proofs) will be strictly limited to (1) errors in author names or affiliations, (2) errors of scientific fact that would cause misunderstandings to readers, and (3) printer's (introduced) errors. Please return the copyedited file within 2 business days in order to ensure timely delivery of the PDF proof. 

If you are likely to be away when either this document or the proof is sent, please ensure we have contact information of a second person, as we will need you to respond quickly at each point. Given the disruptions resulting from the ongoing COVID-19 pandemic, there may be delays in the production process. We apologise in advance for any inconvenience caused and will do our best to minimize impact as far as possible.

EARLY VERSION

PRESS

PROFILE INFORMATION

Thank you again for submitting the manuscript to PLOS Medicine. We look forward to publishing it. 

Best wishes, 

Richard Turner, PhD

Senior Editor 

PLOS Medicine

plosmedicine.org